# Fluorescence Microscopy Methods for the Analysis and Characterization of Lignin

**DOI:** 10.3390/polym14050961

**Published:** 2022-02-28

**Authors:** Agustín Maceda, Teresa Terrazas

**Affiliations:** 1Laboratorio Nacional de Investigación y Servicio Agroalimentario y Forestal, Universidad Autónoma Chapingo, Texcoco 56230, Mexico; biologoagustin@hotmail.com; 2Instituto de Biología, Universidad Nacional Autónoma de México, Mexico City 09230, Mexico

**Keywords:** fluorescence microscopy, lignin, epifluorescence, confocal laser scanning microscopy, fluorescence analytical methods

## Abstract

Lignin is one of the most studied and analyzed materials due to its importance in cell structure and in lignocellulosic biomass. Because lignin exhibits autofluorescence, methods have been developed that allow it to be analyzed and characterized directly in plant tissue and in samples of lignocellulose fibers. Compared to destructive and costly analytical techniques, fluorescence microscopy presents suitable alternatives for the analysis of lignin autofluorescence. Therefore, this review article analyzes the different methods that exist and that have focused specifically on the study of lignin because with the revised methods, lignin is characterized efficiently and in a short time. The existing qualitative methods are Epifluorescence and Confocal Laser Scanning Microscopy; however, other semi-qualitative methods have been developed that allow fluorescence measurements and to quantify the differences in the structural composition of lignin. The methods are fluorescence lifetime spectroscopy, two-photon microscopy, Föster resonance energy transfer, fluorescence recovery after photobleaching, total internal reflection fluorescence, and stimulated emission depletion. With these methods, it is possible to analyze the transport and polymerization of lignin monomers, distribution of lignin of the syringyl or guaiacyl type in the tissues of various plant species, and changes in the degradation of wood by pulping and biopulping treatments as well as identify the purity of cellulose nanofibers though lignocellulosic biomass.

## 1. Introduction

Lignin is one of the main components of the plant cell wall, as it provides structural rigidity to withstand differences in water pressure in vascular tissue [1]; in addition, lignin accumulates in specialized cells for support and storage, such as fibers, sclerenchyma, and parenchyma [2,3], and in the epidermis and cortex in conjunction with suberin and cutin [4]. Cellulose and hemicelluloses are part of the cell wall. Cellulose is the principal component with an accumulation of 40–44% [5], mainly in the form of fibrils packed in crystalline or amorphous form; the higher the percentage of crystalline cellulose, the greater the hardness of the cellulose [6]. The accumulation of hemicelluloses in the cell wall is 15–32%, and its main function is to bind cellulose with lignin; however, hemicellulose are more susceptible to being degraded by the attack of pathogenic fungi [5]. In contrast, lignin is more resistant to pathogens attack and accumulates in the cell wall between 18–35% [5]. Lignin in the cell wall and in the lignocellulosic biomass is a limiting factor in the use of plant biomass because lignin is the restrictive barrier for the penetration of cellulase enzymes used for the degradation of cellulose [7,8]; therefore, lignin is primarily responsible for the recalcitrance of lignocellulose [9,10].

The study of the composition of the cell wall of tree and herbaceous species is due to the potential for use in the paper industry [11,12] and biofuels [13,14]. Currently, there are standardized analytical techniques to characterize the composition of the cell wall, from methods to quantify the percentages of each cell component, such as the TAPPI standards [15], to specialized methods as chromatography, nuclear magnetic resonance, and spectroscopy to analyze and characterize the structural monomers of each lignocellulosic component [16].

Analyzing the structure of lignin in plant tissues and biomass is important because it provides information on the quality of cellulosic materials by identifying the presence of lignin in cellulose nanofibers, which affects the quality of production [17], such as in the identification of potential use of lignin from biofuels to its use as lignin nanoparticles [18,19]. In addition, based on the percentage of lignin present in the biomass, the potential for degradation of celluloses and hemicelluloses for fermentation and biofuel production can be identified [7]; and finally, the chemical composition of lignin allows identifying the possibility of hydrolyzing and using lignin for the production of biofuels [20]. The presence of greater amounts of syringyl (S) than guaiacyl (G) in the lignin structure is one of the key factors for the identification of species susceptible to degradation by pulping, hydrolysis, or enzymes [21]. Syringyl-rich lignin has less condensation, less complex structure, smaller pore size, and higher β-ether content [22,23], while G monomers are linked by β-5, 5-5 and β-β bonds, which are more resistant to degradation [24].

The presence of autofluorescent monomeric structures in lignin is used in different fluorescence microscopy techniques to qualitatively and semi-qualitatively analyze plant cell walls and lignocellulosic biomass. The main advantage of fluorescence microscopy methods is the possibility of observing and analyzing the samples without the need to carry out a staining method in addition to the fact that it is not necessary to degrade or modify the samples as in other analytical methods [25]. There are several methods of fluorescence microscopy to analyze the structural components of plant tissues, living cells, and protein binding. However, the objective of this review is to identify and describe only the fluorescence microscopy methods used for the analysis and characterization of lignin from its polymerization, structure, and degradation to pulping and biopulping of lignocellulosic biomass.

## 2. Structure and Autofluorescence of Lignin

Lignin is a heteropolymer composed of three main monomers, namely *p*-coumaryl, coniferyl, and sinapyl alcohol [26]; once synthesized in the lignin molecule, the name of the monomers are *p*-hydroxyphenyl (H), guaiacyl (G), and syringyl (S), respectively (Figure 1) [27]. However, another type of monomer present in lignin, caffeyl alcohol, has recently been discovered, mainly in the seed coat [28]. Lignin monomers are joined primarily by ether and carbon-carbon bonds, forming the guaiacylglycerol-β aryl ether, phenylcoumarans, diarylpropane, resinol, biphenyl, and diphenyl ether structures [22]. The major bond in lignin is β-O-4, which predominates in syringyl-rich lignin, while lignin with higher guaiacyl content has β-5, β-1, β-β, 5-5, and 5-O-4 bonds [29]. The presence of S-type or G-type lignin is important because the structure of the lignin varies in its conformation. Generally, S-type lignin predominantly has linear chains with fewer cross-links than G-rich lignin due to the fact that lignin S has methoxylated groups, which block the C-5 position of the syringyl units, resulting in few highly stable 5-5 and β-5 bonds [21].

One of the properties of lignin is the presence of autofluorescence generated due to the presence of fluorophores in the lignin structure. Fluorophores are molecules that have the ability to absorb energy at a certain excitation wavelength that causes an excited electronic singlet state; during this short period (few nanoseconds), electron undergoes energy dissipation that is emitted in the form of light when the fluorophore returns to the ground state with a longer emission wavelength [30]. For each photon absorbed, there is a photon emitted, so the intensity of the fluorescence emission is directly proportional to the intensity of excitation, which, over time, causes the photo destruction of the fluorophore (photobleaching). To minimize the photodamage, it is necessary to reduce the exposure time and intensity, prepare the samples, and mount them with anti-quenches/fading agents [31].

For lignin, phenylcoumarans and stilbenes have been identified as the main chromophores [32] (Figure 1) although coniferyl alcohol, biphenyl [33], and to dibenzodioxocins [34] are also considered as lignin chromophores. Fluorescence emission is mainly due to benzene rings present in all lignin monomers. The presence of different side chains on the benzene ring and substitutions at various positions on the ring or side chain are what produce variations in the emission of fluorophores [35]. The blue tones of lignin are due to the grouping of carbonyl groups and the restriction of intramolecular rotation [36]. Therefore, different types of lignin can be identified based on the bands of the fluorescence spectra and also on the shades that are observed by fluorescence microscopy [37]. The lifetime of fluorescence is lower in fluorophores bound to lignin due to the fact that lignin structure presents random and chaotic bonds as well as branches compared to fluorophores bound to other structural components, such as cellulose [38]. In addition, the intensity of fluorescence varies based on pH; the more alkaline (pH 9), the greater the intensity if visible light is used, while for UV excitation, there is no change in intensity [39].

Lignin excitation ranges are given with UV and visible light [40], the first with an emission in bluish tones and the second with an emission that is observed in the spectrum of visible light and includes reddish tones [41]. Because lignin fluorescence spans a broad excitation and emission spectrum, fluorescence lignin has been used for structural characterization within plant tissues, and lignocellulose biomass residues [42], due to degradation treatments, such as high temperatures and application of acids or enzymes, cause the lignin autofluorescence to decrease due to the breaking of the β-aryl-ether bonds so that the re-condensed lignin becomes a dense polymer into electrons, generating new, uncharacterized bonds [43].

Most fluorescence microscopy techniques take advantage of lignin autofluorescence to characterize and analyze lignin. However, biologically active monolignol analogs can also be synthesized with fluorophores, such as dimethyl-aminocoumarin (DMAC) and nitrobenzofuran (NDB), which bind to coniferyl alcohol monomers and that allow to identify the interactions between proteins and monolignols during the process of transport of monomers through the cytoplasm for their polymerization in the cell wall [44].

## 3. Fluorescence Microscopy Methods

Brightfield microscopy is the predecessor of fluorescence microscopy since for the analysis of lignin in plant tissues, methods such as safranin-fast green staining are used for the detection of lignified and non-lignified vascular tissue; the first will be observed in reddish tones, while the second will have greenish tones [45,46]. With the Mäule stain, it is possible to distinguish between lignin S and lignin G since the former has a red hue, while the latter has brown tones. Another method used to detect the presence of lignin is that of fluoroglucinol (Wiesner stain), which binds with the terminal 4-O bonds of *p*-hydroxy-cinnamaldehydes and *p*-hydroxy-benzaldehydes [47,48,49]. In addition, with tetramethylbenzidine (TMB), the activity of peroxidases is detected, which are key for the polymerization of lignin in the plant cell wall [49,50].

In the case of fluorescence microscopy, there are several methods for lignin analysis, such as those used to obtain qualitative images: epifluorescence microscopy (Epi) and confocal laser scanning microscopy (CLSM). In addition, there are methods that allow semi-quantitative analysis, such as fluorescence lifetime imaging microscopy (FLIM), total internal fluorescence reflection (TIRF), fluorescence recovery after photobleaching (FRAP), Föster resonance energy transfer (FRET), microscopy of two-photon excitation (TPM), and simulated emission depletion (STED). Various dyes can be used to stain lignified tissues, such as acriflavine, safranin, and berberine sulfate [51,52,53], which preserve the emission of fluorescence for longer and avoid photobleaching of the sample since prolonged exposure to excitation by UV, white light, or laser can cause the fluorochromes to lose the ability to emit fluorescence [54]. When preparing the samples, the medium in which they are mounted and the pH must be considered since this can alter the intensity of fluorescence [39]. All fluorescence microscopy techniques provide valuable information on their own although they can complement each other or in conjunction with other analytical techniques, such as spectroscopy or chromatography.

A characteristic that the different techniques share is that they work at similar wavelengths both to excite the sample and to detect their fluorescence. In general, the range goes from 400 to 640 nm. However, some methods, such as excitation, use UV ranges from 300 nm of excitation, while some lasers reach up to 820 nm of excitation [55]. For each technique, the power of the lasers or lamps and the type of filter cube that discriminates the wavelength that is emitted or detected vary.

Based on the intensity and wavelength in which the compounds are detected, whether in the blue, green, or red channel, some of the structural components present in the samples can be identified. In the case of lignin, the range in which fluorophores emit fluorescence is wide, so their presence can be detected in different channels [40] depending on the wavelength in which fluorophores are excited and detected in addition to the treatments or dyes that the sample receives, so the detection value can vary based on the method used [55].

### 3.1. Wield-Field Fluorescence or Epifluorescence (Epi)

Epi involves the simultaneous illumination and detection of the entire field of view with low doses of photons and the rapid acquisition of the image to avoid photobleaching. However, the main disadvantage is that Epi collects quite out-of- focus light (Figure 2), so a deconvolution procedure can be applied to the image to reduce the amount of out-of-focus light and obtain a better-quality image [56]. With Epi, is possible to analyze anatomical samples of plants with different thicknesses. Taking the image depends more on the preparation of the tissue than on its thickness, so an adequate preparation technique must be carried out both to minimize eliminate the autofluorescence of unwanted compounds as well as to highlighting the elements to be analyzed [57].

For the study of lignin, with Epi, lignin is detected by autofluorescence, so the distribution of lignin in phloem fibers of species important in the fiber industry can be analyzed [58]. Moreover, Epi detects differences in the percentage of lignin present in the plant tissue due to the intensity of fluorescence [39]. In addition, analyses of the quality of the wood can be carried out [59] to detect changes in the structure of lignified wood after thermal treatments [60].

Furthermore, with Epi, observations of stained samples are made. For example, with the Mäule procedure, analyses can be carried out that allow identifying differences in the composition of lignin based on the presence of syringyl or guaiacyl monomers [61]. With safranin-fast green staining, shade differences in the secondary xylem of various cacti species are detected due to the presence of lignin-rich S or G monomers [62]. Based on different types of dyes, such as Congo red or calcoflour, it is possible to make observations and take images of cellulosic structures and, in combination with lignin autofluorescence, determine the distribution of lignin in different timber species [57]. In addition, staining with safranin alone identifies the distribution of lignin [63], such as effects on degradation of lignified walls through brown-rot fungi attack [64].

### 3.2. Confocal Laser Scanning Microscopy (CLSM)

Similar to Epi, CLSM uses a focused laser at a defined point and at a specific depth and performs a transverse and axial scan to collect all the emitted fluorescence information by a point detector consisting of a pinhole. The pinhole eliminates most of the light outside the focal plane, so better-quality images can be obtained compared to Epi [65] (Figure 3).

The use of this method precedes and is the basis of other techniques because the image is first taken of the area to be analyzed, and later, analyses are carried out with techniques such as FRET [66,67]. CLSM is used in the observation of autofluorescence of plant tissues to calculate the relative amount of lignin present in the plant tissue of *Arabidopsis thaliana* [68,69] and characterize and identify the distribution of lignin in compression and in normal wood of *Pinus radiata* [70]. New staining methods can also be tested to detect the quality of lignocellulosic biomass [71]; determine optimal tissue pH conditions to enhance lignin fluorescence emission [39]; analyze the changes in the structure of healthy, chlorotic, and degraded pine leaves [41]; and study the changes due to degradation of wood two thousand years old with respect to current wood [72]. Fluorescence through dyes, such as Congo red or fluorol yellow, in conjunction with lignin autofluorescence is also used to identify differences in the distribution of lignin in plant tissues of different species [57]. These differences in distribution can also be detected by using dyes for lignin [73], such as basic fuchsin [74] or safranin [53].

On the other hand, CLSM is a suitable method to verify the purity of lignin-free cellulose nanofibers [75]. In addition, CLSM is used to analyze biopulping procedures with lignin-degrading fungi [76], study the effectiveness of pretreatments for the degradation of recalcitrant biomass [77,78], and investigate the effectiveness of lignin-degrading peroxidases whose importance lies in the fact that they can be used for the purification of cellulose and the use of lignin residues [79,80]. CLSM is also used to test the efficiency of fluorescent synthetic monolignols that serve to identify the transport and polymerization in the cell wall of lignin in *Arabidopsis* [81,82,83,84], *Pinus radiata* [85], and *Linum usitatissmum* [86]; determine the polymerization of lignin with laccases [87]; and obtain the distribution of lignin in plant tissues [88].

### 3.3. Fluorescence Lifetime Spectroscopy (FLIM)

FLIM is a technique that makes it possible to identify the time that elapses between the excitation of the fluorophore, the emission of fluorescence, and its decay, which is measured in picoseconds or nanoseconds, and values obtained can be statistically analyzed [89]. However, FLIM analysis can only be used to infer the chemical changes in the sample but does not give information on what causes the changes, which may even be due to changes in the environment, such as pH [90]. FLIM measurement is done by single photon or multiple photons [40]; the latter has the ability to analyze up to 80 μm deep, but unfortunately, with multiple photons, the cellular structure of the samples can be damaged. However, FLIM can be sensitive to various internal factors, such as the structure of the fluorophore, and external factors, such as temperature, polarity, and the presence of fluorescence quenchers [91]. The fluorescence lifetime of loosely packed lignin is on average 4 ns and is associated with secondary walls, while densely packed lignin has a shorter lifetime, between 0.5 to 1.0 ns, so the difference in fluorescence lifetime of the analyzed samples provides relevant information about the type of lignin being analyzed [25].

With FLIM, the lignin fluorescence and the monomeric structures of lignin are detected since differences have been detected in the emission time and intensity of fluorescence due to the presence of different structural monomers when combining FLIM analysis with techniques such as spectroscopy Raman [90]. In addition, by combining CLSM to detect the intensity of fluorescence emission, in conjunction with FLIM, the distribution of different fluorophores with different lifetimes can be observed directly in plant tissues, which allows identifying variations in the composition of compression wood and normal wood [89].

Another analysis that can be performed is the detection of changes in the structural composition of lignin in samples treated with H_2_SO_4_ and NaOH, for which changes in the fluorescence decay time are observed between untreated fibers with respect to the treated fibers [92]. These differences are also observed in samples treated by hydrolysis [93]. Therefore, some authors suggest that with this technique the saccharification rate can be efficiently calculated by means of hydrolysis, acid, and enzyme treatments without resorting to costly analytical methods [43]. On the other hand, using FLIM makes it possible to detect the efficacy of fungal degradation by comparing autofluorescence lifetimes between healthy and infected wood [94]. FLIM is also a suitable method to identify the purity of cellulose nanofibers by determining the absence or presence of lignin [38].

Some authors use FLIM to identify species, such as walnut, beech, spruce, and maple, through the differences in the lifetime of the fluorescence without the need for destructive procedures [95]. Escamez et al. [96] proposed that with FLIM, it is possible to obtain information on the S/G ratio by directly analyzing the vascular tissue of the plants because differences were obtained in the fluorescence lifetimes of the vascular tissue of *Arabidopsis thaliana* based on the distribution of S and G [69].

A variant of FLIM is fluorescence spectral and lifetime measurement (SLiM), proposed by Terryn et al. to analyze the interaction between lignin-degrading enzymes and lignocellulose directly in plant tissues [97]. The difference with respect to FLIM is that it obtains the fluorescence lifetime and the CLSM image in a single wavelength range, with SLiM the range from 455 nm to 655 nm divided into 16 spectral channels so that both the images as the fluorescence lifetime are obtained for each channel. Subsequently, the lifetime fluorescence of each spectral channel is plotted, and the differences can be observed and compared specifically for each spectral.

### 3.4. Two-Photon Microscopy (TPM)

With this technique, thick samples are analyzed, so 3D images can be generated. However, unlike the confocal microscope, there is no photobleaching or phototoxicity above or below the plane in focus, which is why TPM has been used to characterize lignocellulose biomass samples [92]. TPM uses a double excitation by means of two photons simultaneously (Figure 4). Each photon has half the energy of the single-photon excitation event, so the energy of a photon is inversely proportional to its wavelength. In a two-photon excitation, the photons must have a wavelength approximately twice the wavelength of the photons needed to achieve an equivalent transition under one photon exposure. Therefore, if a fluorophore is excited at 400 nm under conventional excitation, fluorophore must be excited by two simultaneous photons at 800 nm; thus, the fluorescence emission after double-photon excitation is similar to that obtained with a normal excitation of a photon [98].

This method is used to analyze autofluorescent and non-fluorescent samples, such as cellulose, since it is based on the signal generated by second-harmonic generation [99], without the need for staining [100]. This type of microscopy has been used to analyze the structural morphology of sugarcane bagasse before and after a hydrolysis and bleaching treatment because when penetrating the sample, better-quality details of the structure are obtained compared to confocal microscopy [93]. Similarly, the recalcitrant structures of lignin were identified in samples treated with enzymatic hydrolysis [55]; therefore, TPM is an efficient method to calculate the saccharification rate of lignocellulosic biomass by detecting changes in fluorescence emission [43]. On the other hand, TPM allows to identify structural changes in the wood coming from musical instruments of different ages [101] and serves to structurally analyze chitosan—lignin composite films [102].

### 3.5. Förster Resonance Energy Transfer (FRET)

This method consists of measuring the molecular interaction between two fluorophores by superimposing the emission and excitation spectra [103]. The process occurs when a donor fluorophore and an acceptor fluorophore are within 10 nm, making it possible for a non-radiative transfer of excitation energy from donor to acceptor to occur [31]. When the distance between donor and acceptor increases, the efficiency of FRET decreases to the sixth power of the distance [97]. Measurement with FRET occurs by determining the changes in fluorescence intensity of the donor and acceptor or by the change in fluorescence over the lifetime of the donor in the presence or absence of the acceptor [31,40].

For the study of lignin, FRET uses the autofluorescence of lignin with the presence of a dye that generates fluorescence, and such is the case of lignin and the dye Rhodamine, where lignin is the donor and Rhodamine the receptor [40,97]. FRET can also be complemented with FLIM by measuring the fluorescence lifetime of the donor alone and of the donor in the presence of the acceptor so that a quantification of the efficiency of FRET and the distance between the donor and the acceptor can be made regardless of fluorophore concentration [31].

FRET has been used to determine the rate of saccharification of species with potential use [25]; in addition to that, FRET has allowed to determine the route of transport and polymerization of lignin in the cell wall [44] and the inhibition of peroxidase and oxidase enzymes due to their association with lignin nanoparticles [66]. On the other hand, FRET has been used as a basis for other methods, such as FRAP, TPM, and the SFLiM variant for the analysis of enzymatic interaction with lignin [103], or in conjunction with CLSM for the detection of nanostructural interactions between additives, such as polyethylene glycol (PEG), for the detection of recalcitrant lignin [67].

### 3.6. Fluorescence Recovery after Photobleaching (FRAP)

FRAP consists of three steps: The first is to mark a region of interest (ROI) in the sample and record the fluorescence intensity prior to photobleaching; then, the ROI is photobleached with a high-power laser beam so that the fluorophores in ROIs are destroyed and irreversibly stop fluorescing (Figure 5). Subsequently, the area around the fluorescent molecules that can freely distribute in the ROI increases, and thus, the ROI fluorescence increases so that the fluorescence recovery reaches a plateau. The scanning fraction that is exchanged between the unbleached area and the bleached area is called the mobile fraction, while the fraction that cannot be exchanged is called the immobile fraction. What is measured is the intensity of fluorescence and the time it takes to recover after photobleaching [31].

Therefore, FRAP has been used to analyze the distribution of lignin-polymerizing enzymes, laccases, and peroxidases and to understand the roles they play in lignin polymerization and lignin distribution in the secondary and primary walls of vascular tissue [104]. In addition, with FRAP the diffusion of macromolecules in the cell wall can be identified [105] and test the diffusion of probes through gels, derived from grasses, that resemble the cellular structure [106]. Moreover, with FRAP, it was possible to determine the saccharification potential of the tissue of *Populus nigra* × *deltoides* [107] by analyzing mobility through their cell walls.

### 3.7. Total Internal Reflection Fluorescence (TIRF)

TIRF uses specific optics to produce illumination light only in the range of 50 to 100 nm at the interface [56], which drastically reduces the light from the bulb and improves the ability to detect fluorescent molecules only on the surface of the sample. TIRF uses the evanescent wave generated when the incident light undergoes total internal reflection or a highly inclined laminated optical sheet (HILO) to illuminate only a partial volume smaller than 200 nm, obtaining the dynamic behavior of a single fluorescent molecule [65] unlike CLSM, which detects the fluorescence emission of the sample from the inner part of the sample. Due to better resolution images, TIRF can be used in conjunction with epifluorescence to characterize lignified fibers and cellular structures [108].

### 3.8. Stimulated Emission Depletion (STED)

STED is a super-resolution microscopy method based on confocal microscopy, in which images are acquired by scanning a point of light focused on a ROI, and fluorescence is collected sequentially pixel by pixel [109]. The system combines blue excitation lasers with red depletion lasers that pass through a phase plate to be patterned into a donut shape in the focal plane. The resulting excitation is a superposition of the two beams, leading to a high-resolution probe scanning the sample, whereby the resulting effective detected fluorescence emission is collected with high spatial and axial resolutions on the nanometer scale, similar to what obtained with transmission electron microscopy (TEM) [110]. However, STED has some limitations because STED can easily damage fluorophores especially in living cells, the cost is high, and the quality may be affected due to red autofluorescence (such as the presence of chlorophyll) [56,111].

The use of STED to characterize the distribution of lignin in plant tissues has been proposed by Päes et al. [109], who mentioned that the STED images to which a deconvolution process was applied show higher quality in regions such as the middle lamina and the secondary wall compared to CLSM. Based on these results, they suggest STED to detect changes in the degradation of wood by enzymatic action.

## 4. Complementary Methods

The methods mentioned above are accompanied by other methods that complement the information collected through fluorescence analysis. Qualitative methods based on transmission and scanning electron microscopy have made it possible to complement the ultrastructural analysis of the lignification of plant tissues [52,76,112] in addition to allowing the characterization of lignocellulosic fibers based on their structure and shape [113,114]. As mentioned in section two, within brightfield microscopy, there are various histological techniques for lignin staining using dyes that can work for both brightfield and fluorescence microscopy, such as basic fuchsin [73] and safranin [62,115].

As for semi-quantitative analytical techniques, the most used in conjunction with CLSM is RAMAN spectroscopy, through which the inelastic dispersion of light is used, and the change in wavelength is detected, which is compiled to obtain information on the excited sample [116]. One of the advantages is that the analysis can be performed at the same time as CLSM in addition to the fact that it is not a destructive technique, and the material can be analyzed directly and without the need for preparation [59,116]. Other spectroscopic techniques are Fourier transform infrared spectroscopy (FTIR) [117] and near infrared spectroscopy (NIR) [118]. FTIR and NIR are based on the infrared absorption with which the molecular vibration of lignin is detected, so the purity of lignin or lignocellulosic materials can be calculated as well as quantifying approximate values of the S/G ratio of purified samples of lignin [15,115].

Within the quantitative techniques to analyze lignin, one of the most used is nuclear magnetic resonance spectroscopy (NMR spectroscopy). By t NMR spectroscopy, lignin of different species are structurally characterized by detecting protons in the structures (^1^H-NMR) or carbon isotopes (^13^C-NMR) and by using two-dimensional heteronuclear single quantum coherence (2D-HQC NMR) that combines the two previous methods and allows structural characterization of lignin [16,119]. Finally, other quantitative techniques less used in the characterization of lignin are chromatographic techniques, such as high-performance liquid chromatography (HPLC) [62,120]. A characteristic of the semiquantitative and quantitative methods mentioned in this section is that they are destructive methods and consume more time and resources. However, NMR spectroscopy and HPLC are necessary to confirm and justify the results obtained with most qualitative analyses, such as microscopy fluorescence.

## 5. Conclusions and Future Perspectives

As we can see in the Appendix A, the most-used method for the analysis of lignin is the CLSM in addition to being used in conjunction with the FLIM, FRET, TPM, and STED techniques; however, Epi in conjunction with FLIM, FRAP, and FRET techniques has been used. Most studies take advantage of the autofluorescence of lignin, and the dyes they use are mainly to dye cellulosic or hemicellulose structures because they do not emit autofluorescence, especially in the FRET method, where lignin and Rhodamine dye interact for the fluorescence emission. On the other hand, although the selected articles were the most recent, the FRET, FRAP, TPM FLIM, STED, and TIRF techniques have recently been applied to analyze the structure and composition of lignin. The data obtained can be semi-quantitative, and they allow comparisons to be made between the samples analyzed quickly, at low cost, and with non-destructive and super-resolution procedures as in the case of STED.

For this reason, with fluorescence microscopy techniques, lignin has been analyzed in a comprehensive manner so that not only the transport and polymerization process in the cell wall is identified, but lignin can also be characterized directly in the vascular or other plant tissues. In addition, the proportion of S/G that allows identifying species with exploitation potential can be calculated, and lignocellulosic biomass can be analyzed to identify changes in its structure, derived from hydrolysis, pulping, and biopulping methods. Finally, although fluorescence microscopy methods are mainly qualitative and semi-quantitative, they are an efficient and rapid tool to characterize lignin.

## Figures and Tables

**Figure 1 polymers-14-00961-f001:**
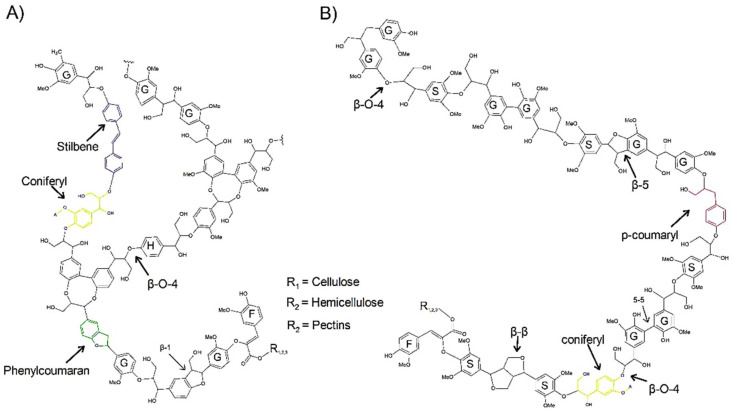
Molecular structure of lignin. (**A**) G-type lignin with the presence of fluorophores. (**B**) S-type lignin with the presence of fluorophores. F, ferulates; H, *p*-hydroxyphenyl; G, guaiacyl; S, syringyl.

**Figure 2 polymers-14-00961-f002:**
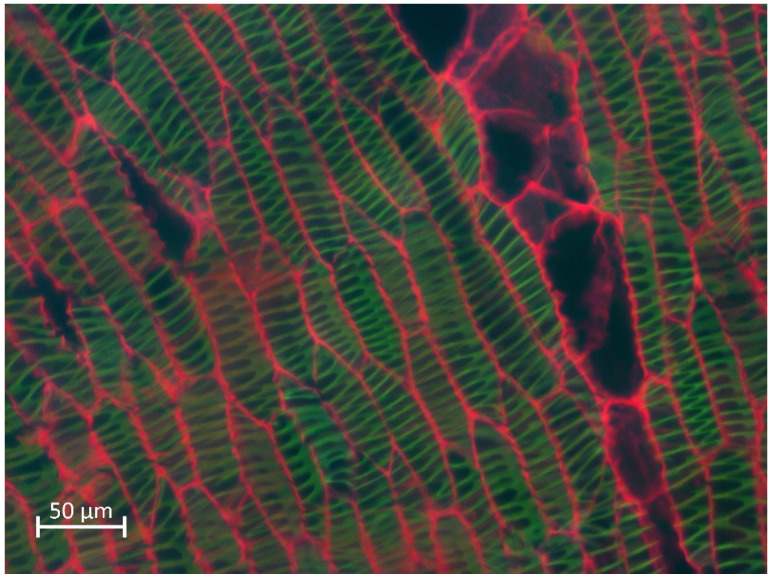
Epifluorescence image from a *Lophocereus marginatus* secondary xylem stained with safranin-fast green, excited at 365 nm (blue), 470 nm (green), and 546 nm (red) at the same time. Scale bar: 20 µm. Image taken from personal file.

**Figure 3 polymers-14-00961-f003:**
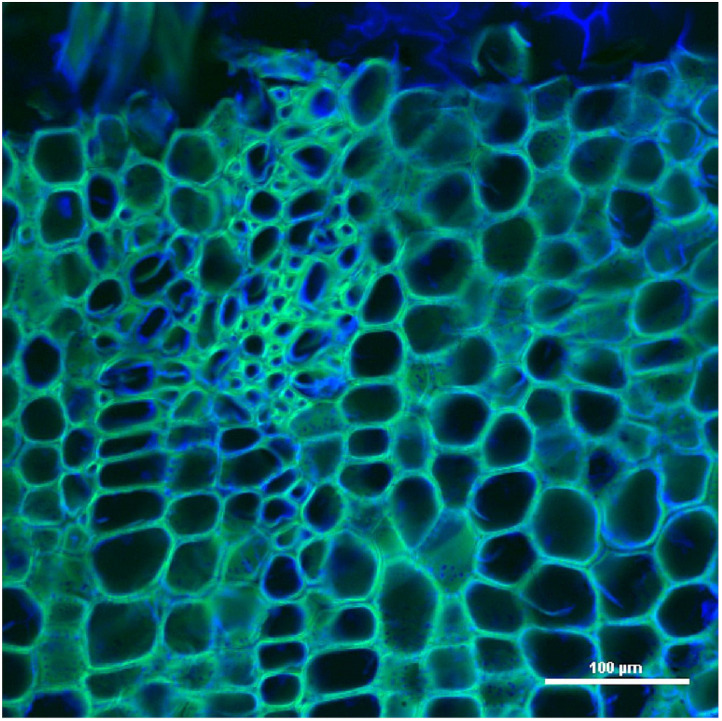
CLSM image from *Cylindropuntia imbricata* secondary xylem stained with safranin-fast green, excited at 405 nm (blue) and 488 nm (green) at the same time. Scale bar: 100 µm. Image taken from personal file.

**Figure 4 polymers-14-00961-f004:**
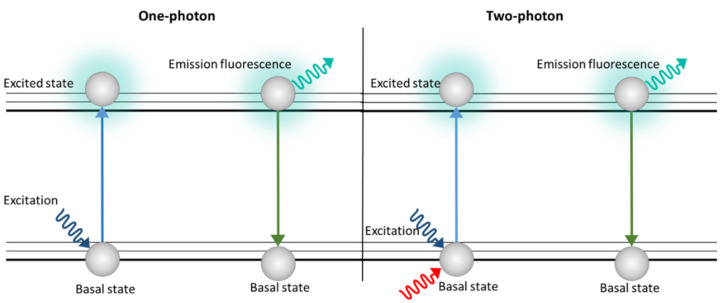
Schematic image of excitation with one photon (Epi, CLSM) and two photons (TPM).

**Figure 5 polymers-14-00961-f005:**
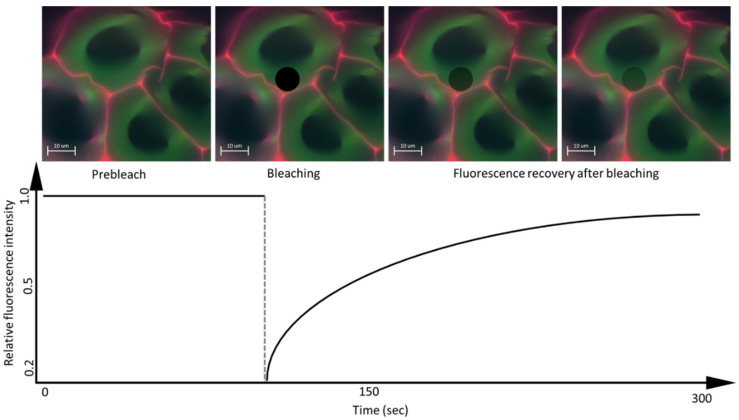
Schematic image of the FRAP method. Image taken from personal file, graphic based in [31].

## Data Availability

Not applicable.

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
