# Peer review of "Fluorescence Microscopy Methods for the Analysis and Characterization of Lignin"

_polymers, 2022, doi:10.3390/polym14050961_

Round 1

Reviewer 1 Report

A comprehensive review study of fluorescence microscopy methods for the analysis is presented. Fluorescence microscopy presents suitable alternatives for the analysis of lignin autofluorescence. Therefore the different methods are analysed that exist to date and that have focused specifically on the study of lignin directly in plant tissue and in samples of lignocellulose fibers. Due to these state-of-the-art methods compiling analyses have been carried out of the transport and polymerization of lignin monomers, distribution of lignin of the syringyl or guaiacyl type in the tissues of various plant species, changes in the degradation of wood by pulping and biopulping treatments, and identifying the purity of cellulose nanofibers though lignocellulosic biomass.

The structure of paper is well organized; the Figs. are clearly denominated and described. One appendix exists as well as with clear structure and description.  The review study brings new findings, interesting information and knowledge.

The manuscript contains only small discrepancies of formal character – for more details see coloured the notices directly in text of the manuscript.

Author Response

We appreciate your favorable comments about  our manuscript . The discrepancies you marked in the text were attended and incorporated in the revised manuscript. All changes are marked now in yellow

Reviewer 2 Report

Greetings, Editor thank you for providing me with the opportunity to review the article. I reviewed the article with title ``Fluorescence microscopy methods for the analysis and characterization of lignin``. The article is compiled by professionals who deeply understand the essence of the subject.  The theme of the article is very interesting and promising in the field. Overall, the structure and content of article is acceptable for the Polymers. I am pleased to send you major level comments. The manuscript can be accepted for publication after modification. Please consider these suggestions and comments as listed below.  

  1. The abstract is written well. Abbreviation words are not suitable for keywords.
  2. The introduction must be concise, and it should be delivered on more clear way with directed necessity for the conducted research work.
  3. Introduction section must be written on more quality way, i.e., more up-to-date references addressed.
  4. Research gap should be delivered on more clear way with directed necessity for the conducted research work.
  5. The main objective of the work must be written on the more clear and more concise way at the end of introduction section.
  6. Line 44, please cite another reference with [7]. Please consider this one (i) Preparation and characterization of nano-sized lignin from oil palm (Elaeis guineensis) biomass as a novel emulsifying agent (ii) Utilization of lignocellulosic biomass: A practical journey towards the development of emulsifying agent.
  7. Figure 2,3 quality is not good please provide high resolution image.
  8. Please must put a space between number and unit. Please revise entire text, several spots have this issue.
  9. Section 5 should be renamed by Conclusion and Future perspectives. Conclusion section is missing some perspective related to the future research work, quantify main research findings, highlight relevance of the work with respect to the field aspect. In the present form conclusion is very weird.
  10. English language should be carefully checked and carefully check paper for language typos.

As already mentioned, these are comments to improve the manuscript and not necessarily to down the quality of work, which is very good.

Author Response

See attach file in which detail explanation is given to your comments.

Reviewer 3 Report

The manuscript entitled Fluorescence microscopy methods for the analysis and characterization of lignin submitted to Polymers Journal.

The concept of the manuscript, fits and suitable to publish in Polymers Journal. This manuscript have many lacking points and thus require substantial major revision before its acceptance.

  • As this manuscript is submitted to Polymers so major focus should be on importance of polymer i.e. lignin and their various applications which is missing so need to discuss in detail.
  • Provide a nice graphical abstract representing the overview of the MS with key highlights. For review article it should be compulsory.
  • Abstract looks very general and not informative. In abstract authors should mention should mention the values of results and importance of research work in one or two sentences.
  • In the introduction section, write the novelty of the work and the problem statement clearly. Authors fails to explain the novelty and importance of the proposed research work thus substantial discussion is essential.
  • Application of lignin in nanaoparticles and PHA production authors can refer International journal of biological macromolecules 128, 391-400, 2019; Bioresource technology 325, 124685, 2021.
  • Give details of other techniques used in lignin characterization in tabular form and also compare with fluorescence microscopy methods.
  • Write the practical applications and future research perspectives and challenges by adding a new section before conclusions
  • The conclusion of the study is not discussed with the specific output obtained from the study, it could be modified with precise outcomes with a take home message.
  • English and grammar mistakes are present. The author should check the manuscript by native English Speaker to improve the quality of the manuscript.

Author Response

In the attach file you will find the detail answer to your comments. All changes are marked in yellow in the revised manuscript.

Reviewer 4 Report

Recommendation: Minor revisions needed.

Comments:

The paper by Maceda et al. contributes different fluorescence microscopy methods for the analysis and characterization of lignin. The title and abstract are appropriate for the content of the text. The article gives an interesting scientific perspective on the analysis of lignin autofluorescence.

Some issues should be addressed prior to publication.

  1. Figure 1. The label on the Figure 1 molecule structure is unreadable, please enlarge the font size.
  2. The page number is not correct on every page.
  3. Page 4, Line 140. “In addition, with tetramethylbenzidine (TMB), the activity of peroxidases (TMB) are detected, which are key 140 for the polymerization of lignin in the plant wall [42,43].” I assume TMB represents tetramethylbenzidine, if that is the case, please correct the abbreviation after it.
  4. Page 4, Line 156 & 162. There should be a period at the end of the sentence. Please double check.
  5. Figure 3. Please change the font size of the scale bar.
  6. Page 7-10. You have five sections of 3.3. Please double check.

Author Response

In the attach file you will find the detail answer to your comments.

Round 2

Reviewer 2 Report

I have reviewed again the manuscript and the mostly concerns are well addressed by authors in the revised version. I suggest that the present form of the manuscript can be accepted for publication.

Author Response

Thanks,

Reviewer 3 Report

Authors have not addressed all the comments thus the details of the following comments should be included in the revised manuscript before its acceptance. 

  • Provide a nice graphical abstract representing the overview of the MS with key highlights. For review article it should be compulsory.
  • In abstract authors should mention should mention the importance of teh review article and future research directions
  • Application of lignin in nanoparticles and PHA production authors can refer International journal of biological macromolecules 128, 391-400, 2019; Bioresource technology 325, 124685, 2021. Should be included.
  • Minor spelling mistakes and units style need to correct.

Author Response

We added a graphic abstract,

2) the abstract was modified according to your suggestions.

3) The references suggested  (2) were incorporated in page 2.

4) the complete manuscript was read again to avoid the spelling mistakes.